# Vascular Stem Cells and the Role of B-Raf Kinase in Survival, Proliferation, and Apoptosis

**DOI:** 10.3390/ijms24087483

**Published:** 2023-04-19

**Authors:** Dipali Goyal, Sean W. Limesand, Ravi Goyal

**Affiliations:** School of Animal and Comparative Biomedical Sciences, University of Arizona, Tucson, AZ 85719, USA

**Keywords:** angioblasts, mesenchymal stem cells, pericytes

## Abstract

Neovascularization is an essential process in organismal development and aging. With aging, from fetal to adult life, there is a significant reduction in neovascularization potential. However, the pathways which play a role in increased neovascularization potential during fetal life are unknown. Although several studies proposed the idea of vascular stem cells (VSCs), the identification and essential survival mechanism are still not clear. In the present study, we isolated fetal VSCs from the ovine carotid artery and identified the pathways involved in their survival. We tested the hypothesis that fetal vessels contain a population of VSCs, and that B-Raf kinase is required for their survival. We conducted viability, apoptotic, and cell cycle stage assays on fetal and adult carotid arteries and isolated cells. To determine molecular mechanisms, we conducted RNAseq, PCR, and western blot experiments to characterize them and identify pathways essential for their survival. Results: A stem cell-like population was isolated from fetal carotid arteries grown in serum-free media. The isolated fetal VSCs contained markers for endothelial, smooth muscle, and adventitial cells, and formed a de novo blood vessel ex vivo. A transcriptomic analysis that compared fetal and adult arteries identified pathway enrichment for several kinases, including B-Raf kinase in fetal arteries. Furthermore, we demonstrated that B-Raf- Signal Transducer and Activator of Transcription 3 (STAT3)-Bcl2 is critical for the survival of these cells. Fetal arteries, but not adult arteries, contain VSCs, and B-Raf-STAT3-Bcl2 plays an important role in their survival and proliferation.

## 1. Introduction

Blood vessels are formed by three distinct processes, namely vasculogenesis, arteriogenesis, and angiogenesis. Angiogenesis is the formation of capillaries from preexisting vessels. During arteriogenesis, the capillaries become muscularized and get converted into vessels with all three layers [1]. However, vasculogenesis is the de novo creation of new vessels from the vascular stem cells (VSC) [2]. Most vasculogenesis occurs during fetal life, and this capacity is significantly reduced in the adult organism [3]. This reduction is evident in each organ, from skin wounds (manifested as non-healing skin ulcers), the heart (in the case of myocardial infarction, poor prognosis in elderly as compared to a young individual), and brain (poor prognosis following a stroke or traumatic brain injury) [4]. The pathways that provide the fetal vessels with increased neovascularization potential are undefined. Of further importance, there is an intensive effort directed toward the ex vivo development of tissues and organs. Similarly, there is a significant effort to implant cells differentiated from stem cells to provide appropriate healing following a stroke or myocardial infarction. Inadequate vascularization of the newly implanted tissue (neovascularization) is a critical factor in the failure of the implants and tissue engineering. Therefore, it is important to understand the pathways involved in blood vessel growth and regeneration.

At present, it is believed that there is a population of VSCs in vessel walls [5,6,7], and that it has markers for mesenchymal stem cells such as CD34 and CD146. These cells are crucial for neovascularization and the formation of a new vessel with all three vessel layers. However, controversy remains regarding the identity of VSC [5], primarily because the current literature does not demonstrate the de novo formation of vessels from isolated VSCs. Additionally, unlike sprouting angiogenesis, the VSCs should lead to neovascularization with the formation of all three layers (adventitia, media, and intima) of vessels. The identity of such VSCs is not well established. Although a population of vascular stromal cells has been isolated from adipose tissue and has been shown to possess mesenchymal stem cells-like properties [8], evidence of such cells in blood vessels is lacking. Also, the pathways responsible for the survival of these VSCs are unknown. Thus, we undertook a study to determine the existence of VSCs and signaling pathways that play an essential role in their survival and proliferation. The identity of VSCs is important in order to move the field forward and is essential for tissue engineering and regenerative biology.

## 2. Results

### 2.1. Fetal Arteries Grow in Serum-Free Media

Cell and capillary outgrowth occurred from the segments of carotid arteries from near-term ovine fetuses in serum-free media, whereas no cell migration or growth was observed from the segments of adult carotid arteries (Figure 1). With the addition of 10% FBS (Fetal Bovine Serum), we observed capillaries and cell growth from both fetus and adult carotid artery segments. Cells from fetal carotid segments grew at a faster rate than adult cells. Thus, we conclude that fetal cells can secrete their own growth factors and do not need supplemental support, whereas adult arterial segments require a supplement of growth factors in the form of FBS to grow.

### 2.2. De Novo Vessel Formation from VSCs

Cells growing from fetal arteries without serum were isolated and plated on 1% collagen gel to form de novo vessels with smooth muscle markers and endothelial markers (Figure 2A–C). These vessels from serum-free fetal cells stained positively for smooth muscle cell marker actin (Red), endothelial cell marker–lectin (green), and DAPI nuclear stain (blue). In contrast, following the addition of FBS, the adult arterial segments only demonstrated capillary growth with endothelial staining, and the smooth muscle actin stain was absent (Figure 2D). Thus, it is evident that the cells from fetal arteries can differentiate into smooth muscle or endothelial cells. Notably, these cells were not able to differentiate into mature adipocytes or osteocytes. Following the purification of endothelial cells from adult arterial segments using CD31 and CD34 antibodies [9] we compared the morphology to fetal vascular stem cells (FVSCs). As shown previously [10,11], the endothelial cells demonstrated a cobblestone morphology (Figure 2E) which was different from FVSCs (Figure 2F). Also, the endothelial cells’ tube formation (Figure 2G) appeared thin and morphologically different from the tube-formation from FVSCS (Figure 2A–C).

### 2.3. Characterization of Fetal Cells Growing from Carotid Artery Segments

The outgrowing fetal carotid VSC were positive for alpha-smooth muscle (smooth muscle cell marker), non-muscle myosin (fibroblast cell marker), vimentin (fibroblast cell marker), CD34 (mesenchymal stem cell marker), Nanog (pluripotency marker), VEGF-R2 (endothelial cell marker), and CD146 (mesenchymal stem cell marker) (Figure 3). However, the cells were negative for smooth muscle myosin and SSEA-4. Proteins detected by immunohistochemistry (IHC) in the fetal cells were validated with western immunoblots for alpha-smooth muscle actin, non-muscle myosin, vimentin, CD34, and nanog (Figure 4). Based on the presence of these specific markers, we infer that VSCs reside in the fetal vessel walls, which have the ability to grow without any supplement of growth factors in the form of FBS, and can form a complete vessel-like structure. These VSCs are significantly different from endothelial progenitor cells [5].

### 2.4. Kinase Pathways Are Upregulated in Fetus Arterial Segments

A transcriptomic comparison between fetal and adult arterial segments indicated that Protein Kinase C (PKC), Ras-Raf-Mitogen Activated Protein Kinase (MAPK)- Extracellular signal-Regulated Kinase (ERK), Rho Kinase, Phosphoinositide 3-kinase (PI3K), Protein Kinase B (AKT), and the glycogen synthase kinase 3 (GSK3) pathways were upregulated in fetal arteries compared to adult arteries (Figure 5). A complete list of the genes altered is provided in Appendix A.

### 2.5. BRAF Inhibition Results in Vacuolar Degeneration of Cells

Based on the transcriptomic analysis, we analyzed seven different kinases to examine their role in (1)B-Raf kinase inhibitor—10 µM (SB590885) [12,13], (2) MAPK-ERK1/2 inhibitor—10 µM (U0126) [14], (3) Rho-associated kinase (ROCK) inhibitor—10 µM (Y-27632) [15], (4) PI3 Kinase inhibitor—10 µM LY294002 [16], (5) GSK-3 inhibitor—10 µM (CHIR 99021) [17], (6) p38 MAPK inhibitor—10 µM (SB202190) [18], (7) the c-JUN N-terminal kinase (JNK) inhibitor—10 µM (SP600125) [19]. Out of several kinase inhibitors tested, B-Raf inhibition by SB590885 significantly reduced the number of viable cells (Figure 6A). Moreover, accumulating evidence indicates that B-Raf kinases regulate cell survival via phosphorylating ERK1/2 [20]. Of note, we observed significant cellular death with B-Raf (yellow bar) but not with ERK1/2 inhibition (Red Bar) (Figure 6A). Thus, it appears that B-Raf inhibition-mediated reduction in cell numbers is because of some other downstream signaling pathway.

### 2.6. BRAF Inhibition Results in Increased Cell Apoptosis

Apoptosis was measured with an Annexin V and 7-amino actinomycin D assay. The percentage of live cells was significantly reduced with the B-Raf inhibitor SB590885, whereas the percentages of late apoptotic cells and dead cells were significantly higher in the B-Raf inhibited-cell group (Figure 6B). ERK1/2 inhibition (U-0126) did not lead to significant apoptosis.

### 2.7. B-Raf Inhibition Leads to Vacuolar Cell Death of Fetal VSC

Following B-Raf inhibition, the VSCs developed severe vacuolization as compared to untreated cells (Figure 6C,D). No other inhibitor used (as mentioned in Section 2.5) produced any morphological change in the VSCs, and were similar to the controls (Figure 6C).

### 2.8. B-Raf Inhibition Leads to Cell Cycle Arrest in G1/Go Phase

We examined the effect of B-Raf and ERK inhibition on the cell cycle (Figure 7). There was a significantly greater number of cells in the G1/G0 phase of the cell cycle with both B-Raf and ERK inhibition. Additionally, cells in the S phase and G2/M phase were significantly reduced in the B-Raf and ERK1/2 inhibited groups. Thus, the inhibition of VSCs cell cycle progression is affected by both B-Raf and ERK1/2 pathways, whereas VSC apoptosis is unique to B-Raf inhibition; however, the inhibition of B-Raf or ERK1/2 inhibits cell proliferation.

### 2.9. Downstream Pathways of B-Raf Mediated VSC Cell Viability and Survival

In cancer cells, studies have demonstrated that B-Raf mediates cell death via Signal Transducer and Activator of Transcription 3 (STAT3) [21]. Thus, we conducted an experiment to determine if the inhibition of STAT3 leads to vascular apoptosis similar to B-Raf inhibition. The inhibition of fetal carotid VSCs by 10 uM Stattic (a selective inhibitor of STAT3) [22] resulted in a significant reduction in viable cell numbers (Figure 8A). Moreover, it has been shown that B-Raf kinase phosphorylates STAT3 at serine-727 and tyrosine-705 [21]. Thus, we examined the effect of B-Raf and ERK1/2 inhibition on the phosphorylation of tyrosine 705 (t705) and serine 727 (s727) by western blot analysis. We observed that the inhibition of B-Raf by SB590885 leads to reduced phosphorylation of STAT3 on both t705 and s727 (Figure 8). Furthermore, the inhibition of ERK1/2 by U0126 led to the reduced phosphorylation of STAT3 on t705 but not on s727 (Figure 8). Thus, we infer that the downstream effect of B-Raf, such as cell survival, which is independent of ERK, may be mediated by s727 phosphorylation, and the effect on cell proliferation, which is mediated by both B-Raf and ERK, may be mediated by t705 phosphorylation. Along this line, studies have demonstrated that STAT3 phosphorylation leads to nuclear localization and DNA binding, in addition to playing a crucial role in cell survival [23].

### 2.10. Constitutive Active STAT3 Prevents B-Raf-Induced Apoptosis

To further confirm the involvement of STAT3 in the B-Raf pathway, we attempted to rescue B-Raf inhibition-induced apoptosis by administering a constitutively active STAT3 construct [24]. This particular STAT3 has the substitution of two cysteine residues (C661A and C663N) within the C-terminal loop of the SH2 domain of Stat3, producing a molecule that dimerizes spontaneously, binds to DNA, and activates the transcription [24]. We tested the hypothesis that B-Raf inhibition decreases STAT3 activation, which is essential for VSCs survival. We observed that, in the presence of activated STAT3, B-Raf inhibition mediated VSCs apoptosis was rescued to a significant extent (Figure 9A). Importantly, previous research has demonstrated that in cancer cells that have constitutively active B-Raf (V600E) activates downstream STAT3, which in turn activates Mcl1 transcription, which is an antiapoptotic factor [21]. Thus, we examined if B-Raf inhibition lowers Mcl1 expression. As shown in Figure 9B, unlike cancer cells, we did not observe any reduction in Mcl1 mRNA levels following the inhibition of B-Raf.

### 2.11. B-Raf Inhibition Leads to the Inhibition of Bcl2 Gene Expression

Mcl1 belongs to the Bcl2 family of proteins [25]. Thus, we examined mRNA levels of several members of the Bcl2 family following the inhibition of VSCs by B-Raf. Concentrations of Bcl-XL, Bcl-W, Bax, Bak, and Bad were unchanged. However, Bcl2 mRNA and protein concentrations were lower in VSCs after inhibition by B-Raf (Figure 10). Notably, ERK inhibition with U0126 did not affect Bcl2 protein expression (Figure 9B). The inhibition of Bcl2 with a well-characterized inhibitor HA14-1 [26] increased apoptotic rates in VSCs similar to B-Raf (Figure 10C). Our bioinformatic analysis demonstrated that the Bcl2 promoter (500 bp upstream to TSS) has several binding sites for STAT3 (Appendix A). Additionally, research has demonstrated that STAT3 can directly regulate Bcl2 promoter activation and expression [27]. Thus, we determined whether it was possible to rescue the B-Raf inhibition-induced apoptosis of VSCs using a library of ~300 compounds that act on different kinases and epigenetic targets. We found that PKC activator PDBu [28], DNMT inhibitor RG108 [29], and histone deacetylase inhibitor Sodium Butyrate [30] were able to rescue the B-Raf inhibitor-mediated apoptosis of fetal VSCs (Figure 11). PKC is a kinase that is also known to regulate B-Raf activation. Similarly, DNMT and histone deacetylase inhibitors are epigenetic regulators.

In summary, the results demonstrate the presence of VSC in fetal arteries, which can be grown in cell culture media without serum or the addition of growth factors. Furthermore, these cells can form vessel like structures de novo when plated on 1% collagen. In addition, these vessel-like structures stain both smooth muscle and endothelial cells. These cells have activated B-Raf kinase, the inhibition of which leads to their apoptosis.

## 3. Discussion

The identity of VSC is a mystery, and the present study provides evidence that VSCs can be isolated from fetal carotid arterial segments. The term VSC has been used to define endothelial progenitor cells or pericytes [5]. However, none of these cells can form de-novo vessels, and they have markers for all three: adventitia (Vimentin), myocytes (smooth muscle actin), and endothelium (CD34). The present study demonstrates that VSCs from fetal vessels can form de novo arterial segments on a collagen gel. Of note, the majority of cells in the fetus vessels are in the synthetic phase, whereas in adult vessels, most of the cells are in the terminally differentiated contractile phenotype [31]. There may be a small population of VSCs in adult vessels, but it is difficult to isolate them. Importantly, we show that cells from fetal vessels have the capacity to initiate de novo vascularization by the process of vasculogenesis, which was absent in the cells obtained from adult vessels. We also demonstrate that B-Raf kinase plays an essential role in the survival of these cells. Notably, B-Raf knockout mice are non-viable and die from vascular defects in utero [32]. These knockout mice demonstrated significant apoptosis in the vessels. Furthermore, several other studies point to an important role of B-Raf in vascular pathologies associated with several life-threatening diseases such as cancer, in which neovascularization plays a vital role [33,34,35].

At present, it is accepted that the chief pathway activated by B-Raf is the ERK pathway [36,37]. However, the B-Raf-mediated downstream mechanisms are not clearly understood. Moreover, the results from this study demonstrate that the B-Raf mediated effect on cell viability is not mediated exclusively through ERK signaling. In contrast, both BRAF and ERK1/2 inhibition resulted in the arrest of the cell cycle in the G1/G0 phase. Thus, it appears that B-Raf also acts on other cell processes through some other downstream mechanisms.

B-Raf is known to regulate STAT3 phosphorylation [21]. In the present report, we show that the inhibition of STAT3 dimerization and nuclear translocation by a selective inhibitor Stattic [22] leads to significant fetal vascular cell apoptosis. Similarly, a study demonstrated that STAT3 phosphorylation leads to nuclear localization, and DNA binding also plays a crucial role in cell survival [23]. In the present report, we demonstrate that B-Raf mediates STAT3 phosphorylation at tyrosine 705 and serine 727. Importantly, our data indicate that the B-RAF-mediated phosphorylation of t705 may be mediated by ERK, but s727 is independent of ERK. Of note, we demonstrate that B-Raf-STAT3 may be mediating the effect on VSCs survival by the activation of Bcl2, which is an important antiapoptotic gene. Finally, the present study shows that B-Raf inhibition-mediated apoptosis can be rescued by PKC activation, DNMT inhibition, or the inhibition of histone deacetylase. Further investigation is needed to determine the effect of B-Raf on these pathways.

## 4. Materials and Methods

### 4.1. Experimental Animals and Tissues

All experimental procedures with animals were performed within the regulations of the National Institutes of Health Guide for the Care and Use of Laboratory Animals, and were approved by the Institutional Animal Care and Use Committees of Loma Linda University and the University of Arizona. For these studies, we anesthetized pregnant ewes (*n* = 6) with thiopental sodium (10 mg/kg, i.v.), and anesthesia was maintained with the inhalation of 1% isoflurane in oxygen during the surgery. The fetus was exteriorized by hysterotomy and euthanized with an overdose of Euthasol (pentobarbital sodium 100 mg/Kg and phenytoin sodium 10 mg/kg; Virbac, Ft. Worth, TX, USA). The isolated carotid arteries were cleaned of adipose and connective tissue and endothelium, as previously described [38,39,40]. Ovine carotid arteries were cleaned of the adventitia in a phosphate-free balanced salt solution of the following composition (mM): 126 NaCl; 5 KCl; 10 HEPES; 1 MgCl_2_; 2 CaCl_2_; 10 glucose; pH 7.4 (adjusted with NaOH). We isolated the cells from near-term fetus sheep and 2-year-old female sheep’s carotid arterial segments by growing arteries in a 3D culture.

### 4.2. Carotid Artery VSC Growth and Isolation

Following dissection, fetal and adult carotid arteries were cut into 5 mm rings. These segments were cultured on 1% rat tail collagen (Cat #08-115, EMD Millipore Inc., Billerica, MA, USA) in DMEM cell culture media with and without 10% FBS (ATCC, Manassas, VA, USA). The cells from carotid arterial segments migrated from the cut end of the explants in 2–4 days of culture, and these were collected from the collagen gel by digestion with collagenase H (Sigma-Aldrich, Inc. St. Louis, MO, USA). The collected cells were cultured in DMEM on a collagen matrix supplemented with 10% FBS and 1% penicillin-streptomycin at 37 °C in 5% CO_2_ and room air. The cells were propagated without endothelial growth factors or any other growth factors in regular DMEM with 10% FBS, and non-adherent cells were removed by changing the media daily. In the adherent cells, enrichment of mesenchymal stem cells by using markers was analyzed after every 7 days of culture, which demonstrated >90% enrichment. These markers were also analyzed in simultaneously grown cells obtained from adult ewes’ carotid artery segments.

### 4.3. Immunoblot Procedures

Artery segments or the isolated cells were homogenized with a homogenizer in an ice-cold cell lysis buffer to isolate proteins (Cell Signaling Technology, Danvers, MA, USA), as described previously [41]. We measured protein content using a commercially available assay kit (Bio-Rad Laboratories, Hercules, CA, USA), and bovine serum albumin (BSA) was used as a reference protein standard. Polyacrylamide gel electrophoresis was conducted to separate the proteins by molecular weight. Following separation, a mini–Trans-Blot Electrophoretic Transfer Cell system (Bio-Rad Laboratories) was used to transfer proteins from the gel to a nitrocellulose membrane at 100 V for three h. We blocked the membranes with blocking buffer (LI-COR Biosciences, Lincoln, NE, USA), and then incubated them overnight with subtype-specific primary antibodies (1:500 dilution) followed by the secondary antibodies (LI-COR Biosciences) for 45 min. We used α-smooth actin (Abcam, Inc. Cat #ab5694) as an internal control for equal protein loading, as well as the blocking peptide for each subtype-specific antibody as a negative control. All antibodies and peptides were obtained from Abcam Inc. (Cambridge, MA, USA). The specific catalog numbers are STAT3 #ab68153, STAT3 (phosphor t705) ab267373, STAT3 (phosph s727) #ab32143, non-muscle myosin #ab138498, vimentin #8069, CD34 #ab762, and nanog #ab173368. The protein bands were detected using an Odyssey Infrared Imaging System (LI-COR Biosciences) and analyzed with ImageJ 1.53t software (NIH).

### 4.4. Epifluorescence Imaging Studies

An Evos Fluorescence Microscope (Thermo Scientific, Waltham, MA, USA) was used to measure the immunofluorescence in cells that were stained using standard techniques [31]. Alpha smooth muscle actin and nuclear stain Dapi (4′, 6-Diamindino-2-Phenylindole -di-Lactate) were used to normalize basal protein expression and cell density, respectively [31]. All antibodies were obtained from Abcam Inc. The images were analyzed using ImageJ 1.53t software, and protein expression was measured as fluorescent intensity/unit area normalized to the fluorescent intensity/unit area of the alpha-smooth muscle actin control.

### 4.5. Cell Viability and Proliferation Assay

To examine the role of different kinases in cell survival, we conducted cell viability and proliferation assays following the manufacturer’s protocol (Cat # MCH100102; Millipore Inc.). For examining cell survival, 10^5^ cells were cultured in several 12-well plates. After 24 h, six wells were left untreated (control), six wells were treated with DMSO (vehicle control), and six wells each were treated for 24 h with the following small molecules: (1) B-Raf kinase inhibitor—10 µM SB590885 [12,13], (2) MEK-ERK1/2 inhibitor—10 µM U0126 [14], (3) Rho-associated kinase (ROCK) inhibitor—10 µM Y-27632 [15], (4) PI3 Kinase inhibitor—10 µM LY294002 [16], (5) glycogen synthase kinase 3 (GSK-3) inhibitor—10 µM CHIR 99021 [17], (6) p38 MAPK inhibitor—10 µM SB202190 [18], and (7) c-JUN N-terminal kinase (JNK) inhibitor—10 µM SP600125 [19]. Stained samples were then analyzed on the Muse™ Cell Analyzer. The increase in cell number suggests proliferation, and a decrease in cell number indicates reduced cell viability.

### 4.6. Apoptosis Assay

Apoptosis was measured with the Annexin V and 7-amino actinomycin D assay that was performed according to the manufacturer’s instructions (Cat # MCH100105, Millipore, Hayward, CA, USA). A total of 10^5^ cells were plated in 12-well tissue culture treated plates. After 24 h in culture, six wells were left untreated (control), six wells were treated with DMSO (vehicle control), and six wells each were treated with the following small molecules: the B-Raf kinase inhibitor—10 µM SB590885 or the MEK-ERK1/2 inhibitor. Samples were then analyzed on the Muse™ Cell Analyzer (Millipore Inc.).

### 4.7. Cell-Cycle Assay

The effect of B-Raf and ERK inhibition on cell cycle progression was determined with a cell cycle assay in accordance with the manufacturer’s instructions (Cat #MCH100106, Millipore Inc.). A total of 10^5^ cells were cultured in several 12-well plates. After 24 h, six wells were left untreated (control), six wells were treated with DMSO (vehicle control), and six wells each were treated with the following small molecules: the B-Raf kinase inhibitor—10 µM SB590885 or the MEK-ERK1/2 inhibitor. Samples were then analyzed on the Muse™ Cell Analyzer.

### 4.8. Real-Time PCR

RNA and protein was isolated by an Allprep DNA/RNA Mini Kit according to the manufacturer’s instructions (Qiagen Inc, Valencia, CA Cat #80204). Isolated mRNA was analyzed using a Qubit (Thermo Scientific). Total RNA (1 µg per reaction) was reverse transcribed using a Quantitect reverse transcriptase kit (Qiagen, Valencia, CA, USA). A real-time PCR was performed on a Bio-Rad CFX Real-Time PCR machine using primers designed using Primer3 and a Quantfast Real-Time PCR Kit (Qiagen, Valencia, CA, USA). Relative expression was normalized to 18S RNA, and fold changes were calculated using the ΔΔCt method with the normalization of individual PCR efficiencies.

### 4.9. Transcriptomic Analysis

To examine the genes altered between fetal and adult vessels, we conducted a whole transcriptomic analysis using second-generation RNA sequencing using the University of Arizona Genetic Core Facility. Briefly, fetal and adult carotid artery segments were thawed and homogenized using three pulses of 4–6 s each. Next, 0.2 mL of chloroform was added per 1 mL of Trizol (0.1 mL per sample). The samples were centrifuged for 15 min at 14,000× *g* (4 °C) and the mixture was separated into a lower red phenol-chloroform, interphase, and a colorless upper aqueous solution. The aqueous phase containing RNA was carefully transferred to a new tube to ensure that the proteins and other cell components did not contaminate the RNA. Next, 250 µL of isopropanol was added to the aqueous phase (0.5 mL/mL of Trizol) and incubated for 10 min. The sample was centrifuged for 10 min at 14,000× *g* (4 °C), and the RNA precipitate formed a white gel-like pellet at the bottom of the tube. The supernatant was discarded, and the sample was vortexed briefly before centrifugation for 5 min at 8000× *g* (4 °C). The supernatant was discarded, and the RNA pellet was air-dried for 10 min. Pellets were resuspended in 50 µL of RNase-free water. The RNA was further purified by Zymo Purelink RNA columns. The obtained RNA samples were measured for quantity (ng/mL) and purity on Nanodrop and Qubit (broad-range) according to a 260/280 nucleic acid absorbance ratio before sending them to the University of Arizona Genetics Core Facility for sequencing, where the RNA was further checked for quality and quantity with an Advanced Analytics Fragment Analyzer (High Sensitivity RNA Analysis Kit—Catalog #DNF-491/User Guide DNF-491-2014AUG13), and the quantity was determined with a Qubit RNA quantification kit (Qubit^®^ RNA HS Assay Kit—Catalog #Q32852).

Once the quality and quantity were validated, a library was constructed from samples using a Swift RNA Library Kit (Catalog #R1024/Swift Protocol version 3.0) and a Swift Dual Combinatorial Indexing Kit (Catalog #X8096). Upon constructing the library, the average fragment size in the library was determined with the Advanced Analytics Fragment Analyzer with the High Sensitivity NGS Analysis Kit (Catalog #DNF-486/User Guide DNF-486-2014MAR10). The quantity was evaluated with an Illumina Universal Adaptor-specific qPCR kit, the Kapa Library Quantification Kit for Illumina NGS (Catalog #KK4824/KAPA Library Quantification Technical Guide, AUG2014).

After completing the final library QC, samples were equimolar-pooled and clustered for sequencing on the NextSeq500 machine. The sequencing run was performed using Illumina NextSeq500 run chemistry (NextSeq 500/550 High Output v2 kit 150 cycles, Catalog FC-404-2002).

RNAseq Data Analysis: Sequence data was quality validated for RNAseq analysis using FastQC Version 0.11.9. Sequences with average Phred scores below 34 were discarded. Fully annotated genome indices were generated for sheep (Oar_rambouillet_v1.0), cow (ARS-UCD1.2), mouse (GRCm39), and human (GRCh38.p13) using ENSEMBL (v1.0.103), and aligned with the sequencing data using Salmon [42]. An integrated differential expression and a pathway analysis was conducted with iDep.96 web-based applications [43]. During preprocessing, the genes with less than 20 counts per million in three or more libraries were discarded from further processing, and counts were normalized using regularized log (rlog) transformation. For pathway analysis, genes with an FDR cutoff of 0.05 and a minimum fold change of 1.5 were used. The genes altered were matched with known Kyoto encyclopedia of genes and genomes (KEGG) pathways [44], and a gene set enrichment analysis (GSEA) was conducted as published [45]. A pathway analysis was conducted using Ingenuity Pathway Analysis software.

### 4.10. Transfection of Constitutive Active STAT3

The constitutive active STAT3 construct was obtained from Addgene Inc. Watertown, MA, USA (Cat #13373). The transfection of VSCs was conducted using X-tremeGene HP DNA transfection reagents following the manufacturer’s protocol (Milipore Sigma Inc. Burlington, MA, USA)

### 4.11. Endothelial Cell Isolation and In Vitro Tube Formation Assay

Cells were isolated from adult carotid arterial segments migrated from the cut end of the explants in 2–4 days of culture, and these were collected from the collagen gel by digestion with collagenase H (Sigma-Aldrich Inc.). The endothelial cells were purified by pulldown using CD31 and CD34 antibodies following the published protocol [9]. The collected cells were cultured in DMEM on a collagen matrix supplemented with 10% FBS and 1% penicillin-streptomycin at 37 °C in 5% CO_2_ and room air.

### 4.12. Statistical Analysis

To determine differences between the groups, we analyzed the data using a one-way ANOVA with a post-hoc Bonferroni’s analysis with GraphPad Prism 9.5.1 software (GraphPad Software Inc., San Diego, CA, USA). The hypothesis was accepted at *p* < 0.05, and for each experiment, n equaled five or more independent animals, as noted.

### 4.13. Limitations

The present study did not purify the VSCs using mesenchymal stem cell antibodies. However, previous reports have demonstrated that, following several passages, the plastic adherence property selects the progenitor cells [46]. Furthermore, the cells staining to stem cells marker CD34 and CD146 demonstrated the staining of >90% of cells. Furthermore, we acknowledge the limitation that the population of cells used for the 3D cell culture may have contaminated with differentiated endothelial, smooth muscle, and adventitial cells. The cells isolated from fetal vessels were able to form a 3-D tube-like structure stained with both endothelial and smooth muscle markers, whereas the cell population from adult arteries only formed tube-like structures stained with endothelial stain. Thus, despite these limitations, we propose that fetal vessels contain a population of cells that provide them with the ability to undergo de novo vasculogenesis, which is absent in adult vessels. However, further investigation is required to fully characterize these fetal VSCs.

## 5. Conclusions

The present study demonstrates that the VSCs are present in fetal arteries and possess the ability to grow independent of FBS-supplemented media, and are capable of forming new vessels in vitro. Importantly, the B-Raf-STAT3-Bcl2 pathway is essential for VSCs survival, and the knockout of B-Raf leads to improper vascular development and fetal death. During the past few decades, our understanding of vascular development has increased significantly. It is important to unravel the mechanisms of vascular development because such an understanding may not only help prevent placental, embryonic, and fetal defects, but also enable us to understand the pathways that play a role in vascular pathologies during adult life, such as neointimal thickening and vascular stiffening. Furthermore, these pathways may be utilized to improve neovascularization in regenerative medicine, tissue engineering, and cell implants.

## Figures and Tables

**Figure 1 ijms-24-07483-f001:**
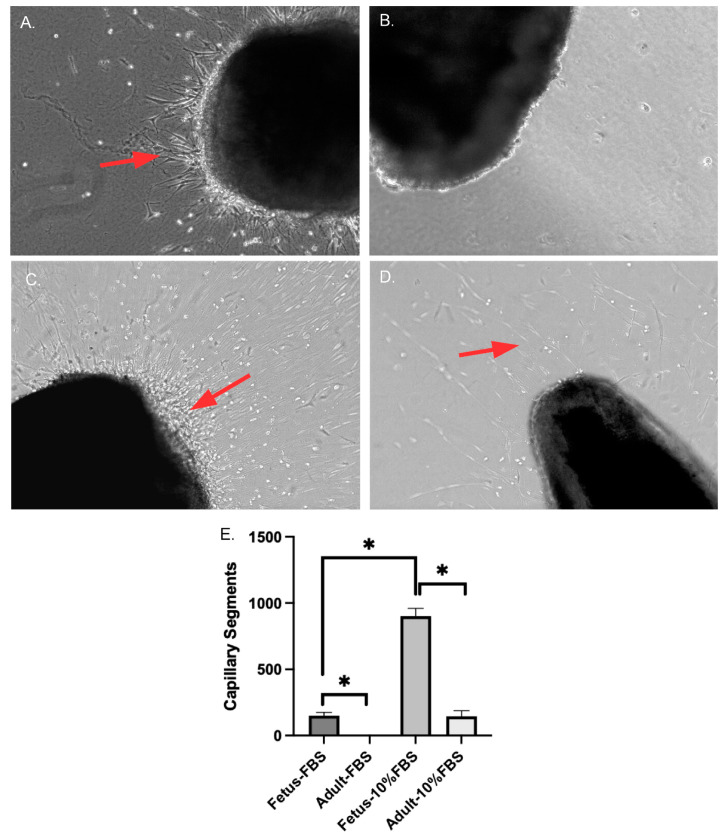
(**A**) Capillary and cell growth from carotid arterial segments from near-term ovine fetus grown in serum-free DMEM media. (**B**) No cell or capillary growths from carotid artery segments from adult ewe in serum-free DMEM media. (**C**) Robust cell and capillary growths in carotid segments from near-term ovine fetus grown in DMEM with 10% FBS. (**D**) Cell and capillary growth from an adult carotid artery segment in DMEM with 10% FBS. (**E**) The bar graph shows a higher amount of capillary segment growth in fetus carotid segments as compared to those from adult sheep. * Denotes *p* < 0.05, *n* = 5. All of the images shown in the figure were taken at 10× magnification.

**Figure 2 ijms-24-07483-f002:**
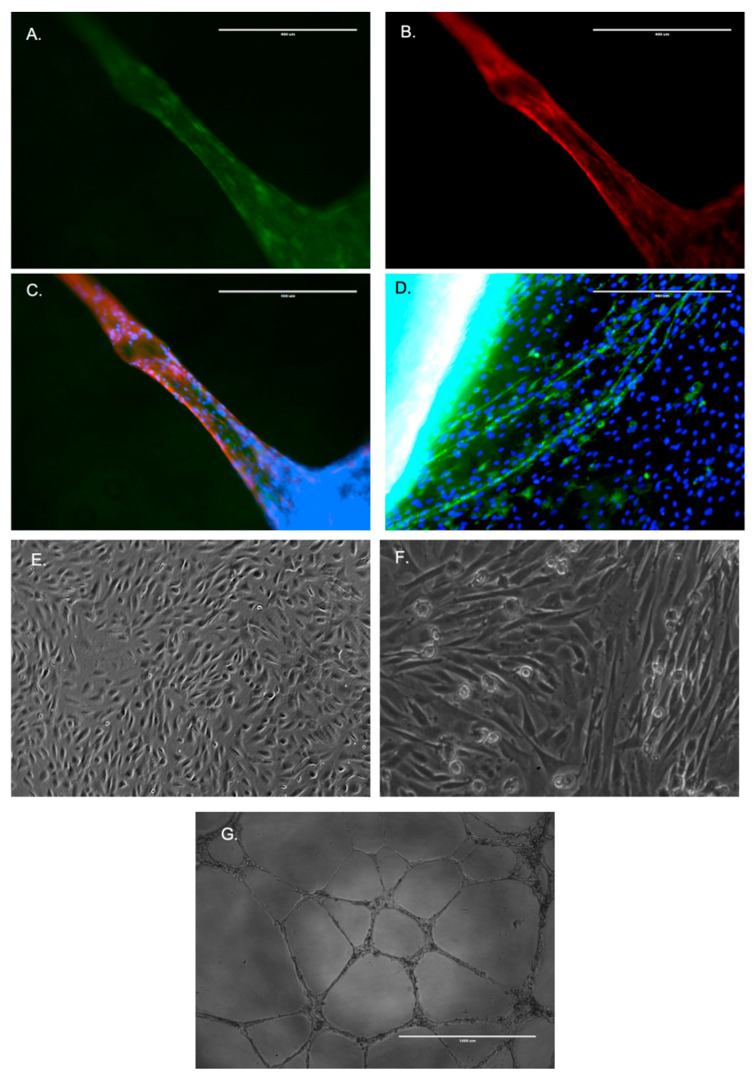
The formation of a new vessel from fetal VSCs. (**A**) The endothelial layer staining by lectin. (**B**) The alpha-smooth muscle actin staining by RFP conjugated antibodies. (**C**) The combined image of smooth muscle cells (red), endothelial cells (green), and cell nuclei (blue) by DAPI live cell stain. (**D**) The staining of sprouting angiogenesis from adult vessel segment and the presence of the endothelial cell layer (green). No alpha-smooth muscle stain (red) was observed. (**E**) The morphology of purified endothelial cells by CD31 antibodies-mediated pull down. (**F**) The morphology of FVSCS. (**G**) The tube formation by purified endothelial cells on a 3D gel matrix. All of the images shown in the figure were taken at 10× magnification.

**Figure 3 ijms-24-07483-f003:**
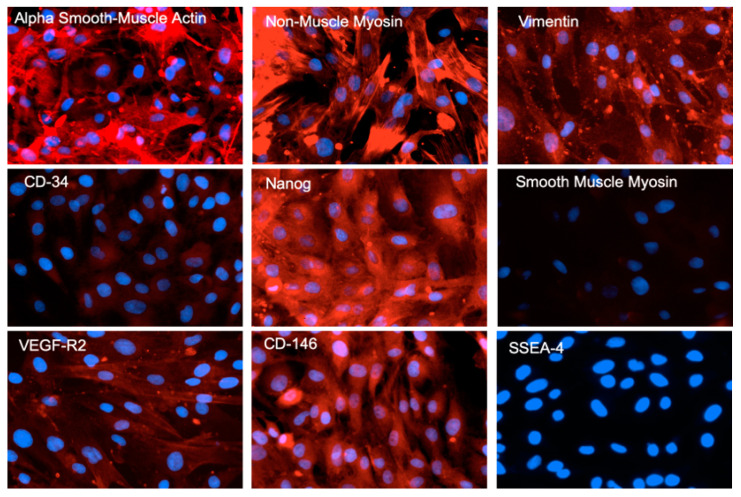
The immune-staining of various markers in fetal VSC. The cell-specific markers were examined using a red fluorescent protein (RFP)-tagged antibody (red fluorescence) against a particular antigen. The red color in Figure 1 indicates the presence of a particular marker, and the blue stain indicates the binding of DAPI dye to the nuclear DNA. All of the images shown in the figure were taken at 10× magnification.

**Figure 4 ijms-24-07483-f004:**
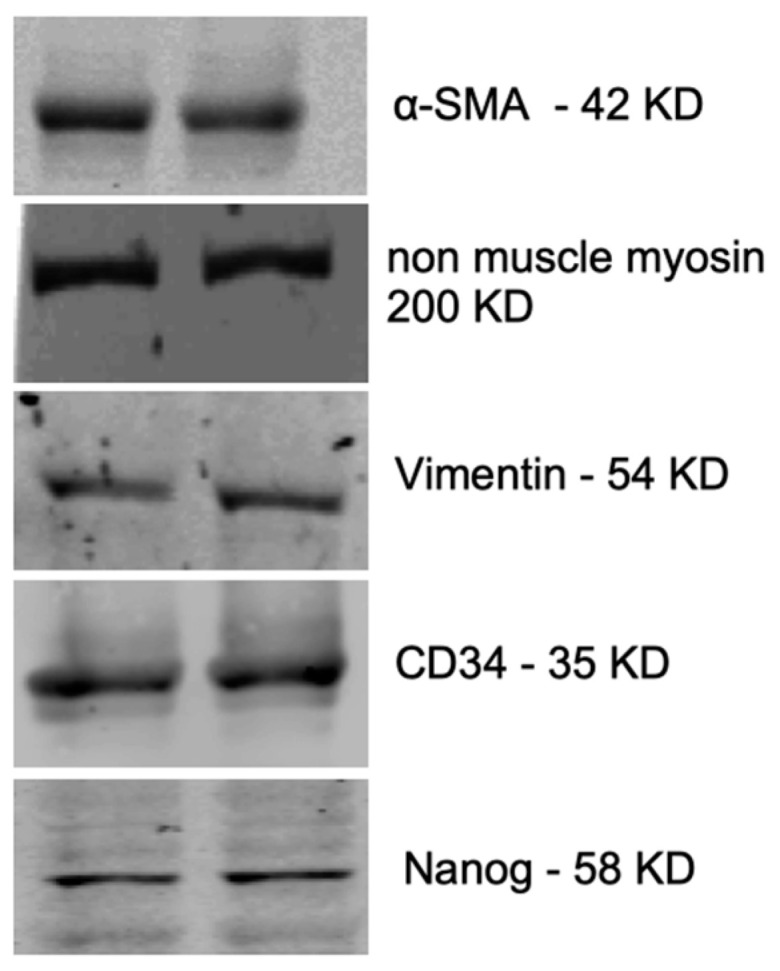
Western blot images of several markers in the fetal VSCs. The two representative bands are from the VSCs from two different ovine fetuses.

**Figure 5 ijms-24-07483-f005:**
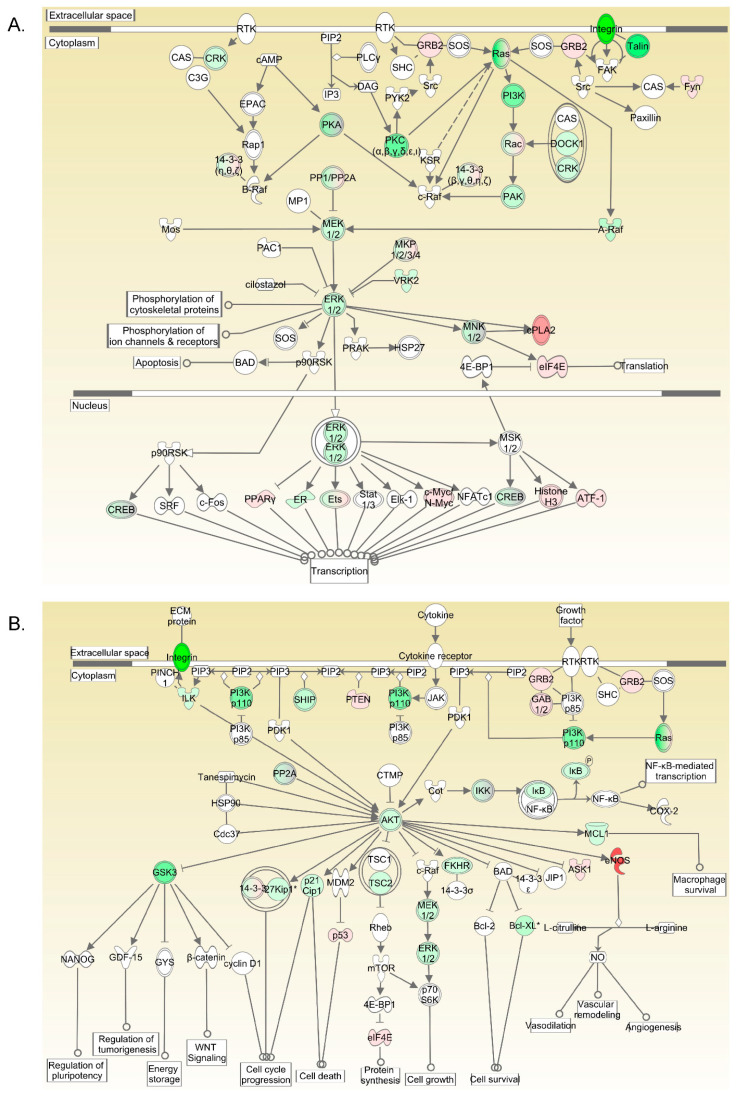
Pathways upregulated in the fetal carotid arteries as compared to those in adults. (**A**) The upregulation of PI3 Kinase, PKC, and ERK1/2. (**B**) The upregulation of GSK3, PI3 Kinase, AKT, and ERK1/2 pathways.

**Figure 6 ijms-24-07483-f006:**
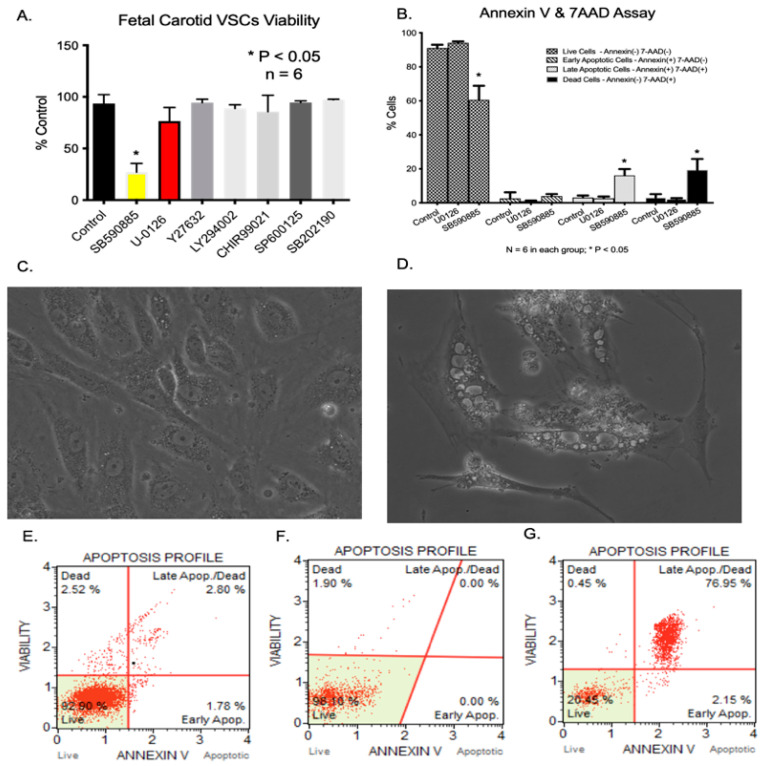
(**A**) Significant reduction in fetal VSCs viability following incubation with B-Raf inhibitor SB590885. There was no significant reduction in VSCs viability in response to other inhibitors. (**B**) Significant apoptosis following B-Raf inhibition. (**C**) Fetal VSCs in DMEM media. (**D**) Cells undergoing vacuolar cell death following incubation with B-Raf inhibitor SB590885 for 24 h. (**E**–**G**) Representative pictures of apoptosis assay following incubation with vehicle control, U1026, and SB590885. * Denotes *p* < 0.05, *n* = 5. Images shown in panels C and D were taken at 20× magnification.

**Figure 7 ijms-24-07483-f007:**
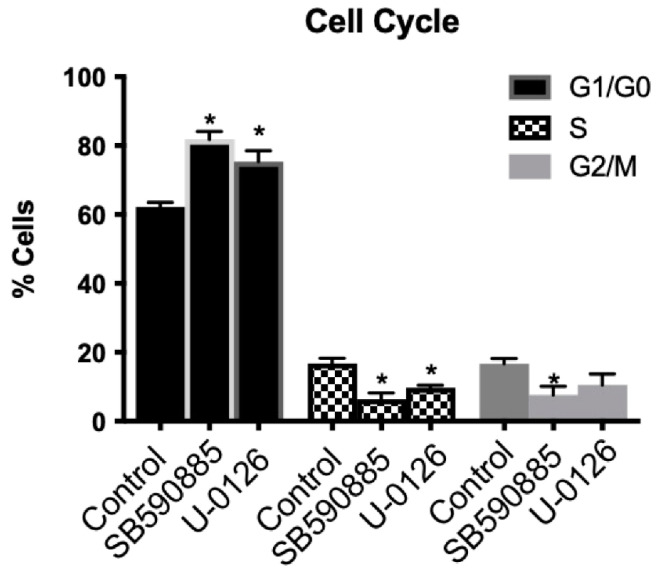
VSCs number in G1/G0 phase (black bars), S-phase (checkered bars), and G2/M phase (grey bars). *n* = 6, and * Denotes *p* < 0.05).

**Figure 8 ijms-24-07483-f008:**
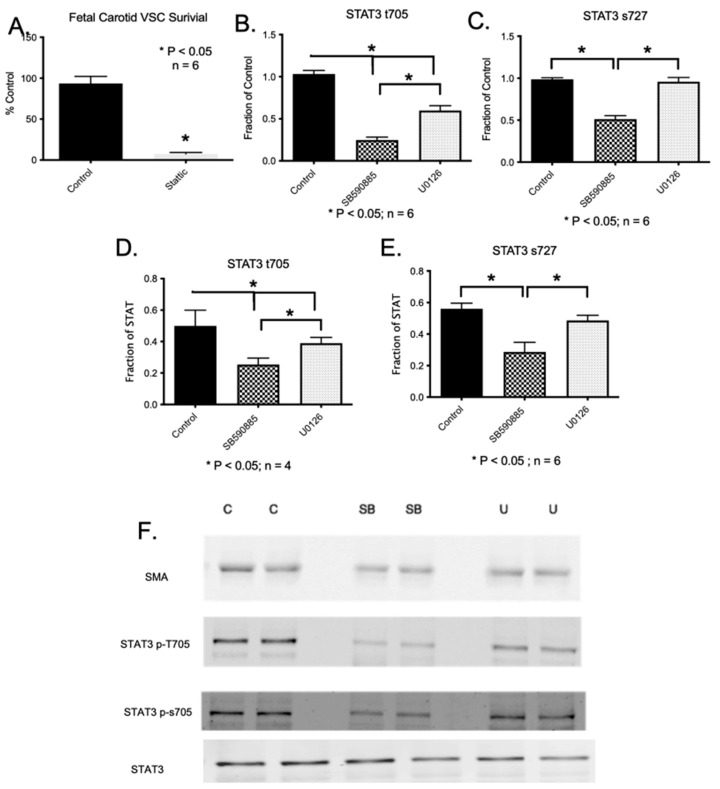
(**A**) A bar graph demonstrates significant VSC death in DMSO (vehicle) control and following incubation with Stattic (STAT3 inhibitor). (**B**) The significant reduction in phosphorylation of STAT3 tyrosine 705 following incubation with B-Raf and ERK inhibitors. (**C**) The significant reduction in phosphorylation of STAT3 serine 727 following incubation with the B-Raf inhibitor. No effect of the ERK inhibitor was observed on s727 phosphorylation. (**D**) The ratio of phosphorylated STAT3t705 and STAT3 Protein. (**E**) The ratio of phosphorylated STAT3s727 and STAT3. (**F**) Representative blots of alpha-smooth muscle actin, phosphorylated STAT3t705, phosphorylated STAT3s727, and STAT3. * Denotes *p* < 0.05, *n* = 5.

**Figure 9 ijms-24-07483-f009:**
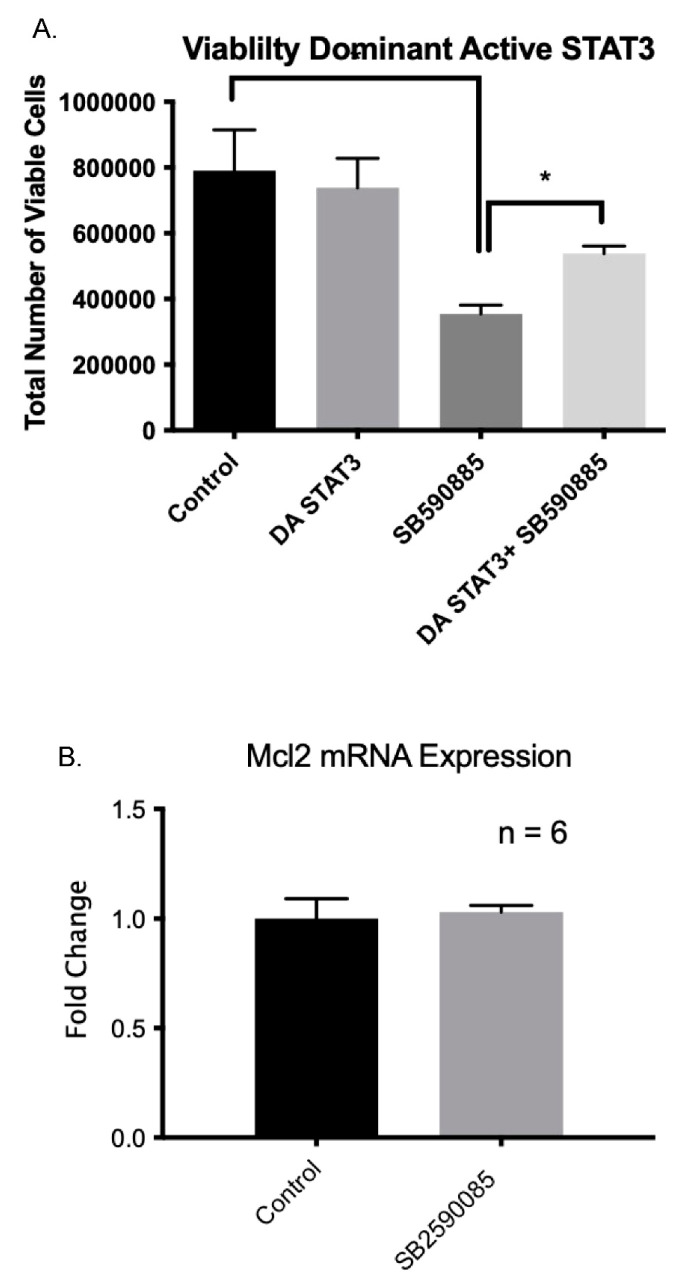
(**A**) The rescue of B-Raf inhibition-mediated apoptosis of VSCs by dominant active STAT3. In addition, dominant active STAT3 itself has no significant effect on VSCs survival. (**B**) No effect of B-Raf inhibition on Mcl1 mRNA levels in VSCs. * Denotes *p* < 0.05, *n* = 5.

**Figure 10 ijms-24-07483-f010:**
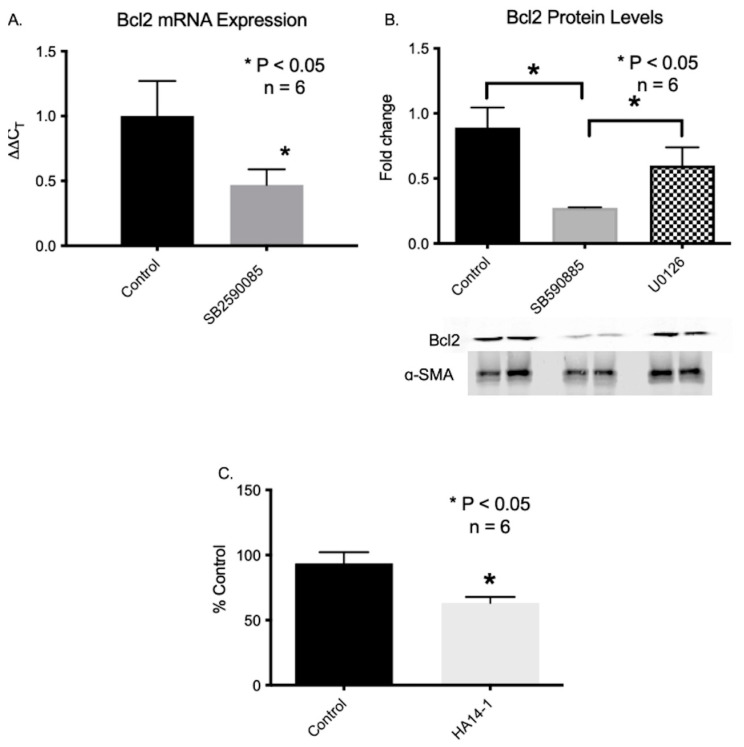
(**A**) The reduction in Bcl2 mRNA following incubation of VSCs with the B-Raf inhibitor. (**B**) The reduction in Bcl2 protein levels in VSCs lysate following incubation with the B-Raf inhibitor. No significant change in Bcl2 protein levels was observed by the incubation of VSCs with the ERK inhibitor. (**C**) The cell death of VSCs following incubation with the Bcl2 inhibitor HA14-1. * Denotes *p* < 0.05, *n* = 5.

**Figure 11 ijms-24-07483-f011:**
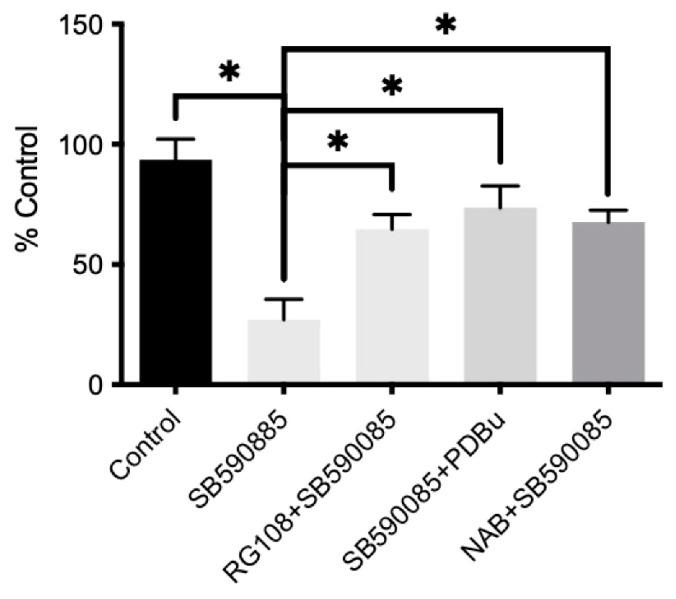
The bar graph demonstrates VSCs viability following incubation with the B-Raf inhibitor. It shows that a DNMT inhibitor (RG108), PKC activator (PDBu), and an HDAC inhibitor (Sodium Butyrate—NAB) were able to rescue B-Raf inhibition-mediated VSCs apoptosis. * Denotes *p* < 0.05, *n* = 5.

## Data Availability

All raw data will be made available upon request.

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
