# Peer review of "Vascular Stem Cells and the Role of B-Raf Kinase in Survival, Proliferation, and Apoptosis"

_ijms, 2023, doi:10.3390/ijms24087483_

Round 1

Reviewer 1 Report (Previous Reviewer 2)

revised

Author Response

Response: Thank you for your valuable time and constructive critique.

Reviewer 2 Report (Previous Reviewer 1)

The heterogenicity of cells obtained by tissue explanted technique and the low percentage of stem cells population in any isolation protocol is addressed by performing an enrichment step that include centrifugation, gentle trypsinization and specific growth media among other possibilities. Once this step is taken, stem cells will outgrow any other cell type that may still be present, as demonstrated in many protocols including the one referenced by the authors. It is not clear to this reviewer whether any such step was taken in this manuscript. The percentage of VSC versus other cell types present in the explanted cells and the enrichment of VSC following culture passage can potentially be addressed by cell sorting and/or flow cytometry analysis. Initially what percentage of cells were positive for CD34 and CD146 and how long it takes for it to reach >90% enrichment? Can the authors provide a temporal figure showing the enrichment of these markers in selected passages (0, 2, 4, 6, … for example) with the concomitant decrease in smooth muscle cells and fibroblasts markers?

Author Response

Comment: The heterogenicity of cells obtained by tissue explanted technique and the low percentage of stem cells population in any isolation protocol is addressed by performing an enrichment step that include centrifugation, gentle trypsinization and specific growth media among other possibilities. Once this step is taken, stem cells will outgrow any other cell type that may still be present, as demonstrated in many protocols including the one referenced by the authors. It is not clear to this reviewer whether any such step was taken in this manuscript. The percentage of VSC versus other cell types present in the explanted cells and the enrichment of VSC following culture passage can potentially be addressed by cell sorting and/or flow cytometry analysis. Initially what percentage of cells were positive for CD34 and CD146 and how long it takes for it to reach >90% enrichment? Can the authors provide a temporal figure showing the enrichment of these markers in selected passages (0, 2, 4, 6, … for example) with the concomitant decrease in smooth muscle cells and fibroblasts markers?

Response: For the isolation of the cells from the fetus's arteries, we did not conduct enrichment with any antibodies. However, we did a comparative study with cells from adult arteries side by side to understand the differences in the cells growing from fetal vs. adult arteries. It took ~7 days for the adherent cells to reach >90% enrichment with CD34 and CD146 positivity. This was not observed in cells from adult vessels. The cells were cultured on a tissue culture-treated flask without a 3D matrix (such as collagen or basal matrix extract). The suspended cells were discarded by washing the flask surface with sterile PBS. In the future, it will be a good idea to enrich the cells with CD34 or CD146 antibodies. We did not do the staining at 0,2,4 and 6 days. After a week of culture, the staining was performed.  These studies were done on near-term fetal sheep and are time-consuming. Staining for the 0, 2, 4, and 6 days cells will take at least another year's worth of studies (sheep gestation is 145 days), and we do not believe it will provide much information.

Reviewer 3 Report (New Reviewer)

In this manuscript, the authors isolated fetal VSCs from the ovine carotid artery, and also validate the functions of VSCs via detecting markers for endothelial, smooth muscle, and adventitial cells. Finally, the authors also identified the pathways involved in the VSCs survival. In general, my assessment of this manuscript is positive, although I believe that it needs to be further improved prior to publication in IJMS.

Specific comments:

1.       In figure 1, 2, 6, the authors should add the missing bar for each image.

2.       In figure 4, the authors showed the expression of several markers in the fetal VSCs, I suggest that the authors should also detect the expression of these markers in adult arteries as control. Additionally, the quality of western blot is not good, please repeat and provide a better western blot image.

3.       In figure 8, I suggest the authors also detect the expression of total STAT3 after DMSO treatment or inhibitor treatment. Then, quantifying the Stat3 t705 or s727/ total Stat3 ratio is better. 

Author Response

Comment: 1.       In figure 1, 2, 6, the authors should add the missing bar for each image.

Response: All the images of Figures 1 and 2 were taken at 10X magnification and Figure 6 at 20X magnification. We have noted this in the figure legends. The pictures were taken by the research technicians and students in the laboratory, and sometimes they miss to check the box to add the scale bar. However, it doesn’t affect the interpretation and conclusions drawn from the images.

Comment: 2.       In figure 4, the authors showed the expression of several markers in the fetal VSCs, I suggest that the authors should also detect the expression of these markers in adult arteries as a control. Additionally, the quality of the western blot is not good, please repeat and provide a better western blot image.

Response: we have uploaded better-quality pictures.

Comment: 3.       In figure 8, I suggest the authors also detect the expression of total STAT3 after DMSO treatment or inhibitor treatment. Then, quantifying the Stat3 t705 or s727/ total Stat3 ratio is better. 

Response: We have included the STAT3 blot and provided the data as the ratio.

Reviewer 4 Report (New Reviewer)

Date:

March 09, 2023

To:

Dipali Goyal

Subject:

Reviewer comments

-       

-      The manuscript (ijms-2289569) entitled “Vascular Stem Cells and the Role of B-Raf Kinase in Survival, 2 Proliferation, and Apoptosis” seems less interesting. initially, the authors studied the neovascularization potential in comparing fetal and adult VSC, but the study focusing only on fetal VSC and finally claiming fetal and adult VSC. There is no clear representation of study direction in the comparison, very few experiments compared both. The experimental designs ambiguous, and the writing must be improved at higher degree at scientific level. The result concluded with deficit in scientific justification. The study has higher scope of citation from the end of stem cell research.

-      The abbreviation needs to elaborate at initial usage.

-      Figure 3 images needs to maintain uniformity in scale (µM) and dapi intensity should be mild than the marker intensity.

-      Supplementary table 1 missing.

-      Figure 5, Image needs to explain well in the point fetal and adult modulation of proteins with clear representation. The figure legend required to explain well for clear understanding. Required separate pathway explanation for fetal and adult.

-      Line 132-136; Mention like B-Raf Kinase inhibitor (SB590885) - 10 μM................ for all inhibitors.

-      The vacuolar images of all the inhibitors for fetal and adult VSC required.

-      Section 2.7; explain deeper at morphological level and correlate with other experimental results. Provide the morphological images of other inhibitors to increase the robustness of data.

-      For cell cycle analysis, the FACs (Muse™ Cell 361 Analyzer) images needs to give along with the bar graph for clear representation.

-      Figure 8D; Include the house keeping protein expression western band and plot the expression intensity graph.

-      Line 219-221; mention reference or provide Bioinformatics data in supplementary images to support the sentence.

-      In method, the fetal VSC cells isolation protocol was missing.

-      The antibody product details like catalogue number required with the dilution ratio used for each method.

-      The reference needs to add most recent research in the field of the vascular generation field.

Overall, the manuscript lacks a clear scientific design and approach and concluded in improperly without proper supporting data. The entire study seems to be descriptive and must improve far better before considering for publication.

Author Response

Comment: The abbreviation needs to elaborate at initial usage.

Response: We have checked the abbreviation and defined them at first instance.

Comment: Figure 3 images needs to maintain uniformity in scale (µM) and dapi intensity should be mild than the marker intensity.

Response: We have made the suggested change.

Comment: Supplementary table 1 is missing.

Response: We have provided Supplementary Table 1.

Comment: Figure 5, Image needs to explain well in the point fetal and adult modulation of proteins with clear representation. The figure legend required to explain well for clear understanding. Required separate pathway explanation for fetal and adult.

Response: We have modified the text to incorporate the suggested changes.

Comment: Line 132-136; Mention like B-Raf Kinase inhibitor (SB590885) - 10 μM................ for all inhibitors.

Response: We have made the suggested change.

Comment: The vacuolar images of all the inhibitors for fetal and adult VSC required.

Response: Only B-Raf inhibition produced the vacuolar images. None other inhibitors produced any morphological change and were similar to the control.

Comment: Section 2.7; explain deeper at morphological level and correlate with other experimental results. Provide the morphological images of other inhibitors to increase the robustness of data.

Response: We have explained that no other inhibitor produced any morphological changes in the VSCs.

Comment: For cell cycle analysis, the FACs (Muse™ Cell 361 Analyzer) images need to give along with the bar graph for clear representation.

Response: We did not keep the images for all experiments. However, we have added the images where available. The raw data of the output other than images is available if needed.

Figure 8D; Include the housekeeping protein expression western band and plot the expression intensity graph.

Response: We have made the suggested change (smooth muscle actin – SMA and total STAT3).

Line 219-221; mention reference or provide Bioinformatics data in supplementary images to support the sentence.

Response: We have added the data as supplemental Figure 1.

In method, the fetal VSC cells isolation protocol was missing.

Response: We have made the suggested change.

The antibody product details like catalogue number required with the dilution ratio used for each method.

Response: We have made the suggested change.

The reference needs to add most recent research in the field of the vascular generation field.

Response: We have referenced recent research.

Round 2

Reviewer 3 Report (New Reviewer)

Accept

Reviewer 4 Report (New Reviewer)

No comments.

This manuscript is a resubmission of an earlier submission. The following is a list of the peer review reports and author responses from that submission.

Round 1

Reviewer 1 Report

This reviewer would like to thank the authors for the responses to the comments and the revised version of the manuscript. The main point of this reviewer’s concerns, however, still stand. How can the authors undeniably determine that cells obtained from the fetal carotid arteries are indeed vascular stem cells (VSC) and not endothelial and vascular smooth muscle cells that migrated? It is well established that structural and functional features of newborn vascular cells change dramatically during the first few weeks of postnatal life, and the differences in fetal vs adult cell growth may reflect that. This reviewer agrees that a VSC population is present, nonetheless in order to conclude that these cells are responsible for neovascularization process warranties thoroughly characterization.

Author Response

The study on fetal arteries was done in comparison with adult arteries. The cells obtained from the fetus were passaged several times and then grown on 3D culture. The adult cells are not only resistant to B-Raf inhibitor cell death but also fail to form vessel-like structures. Again, the adult cells did not stain to various markers as fetal cells. Overall, fetal cells are very different from endothelial cells, as published by us previously {Goyal, 2019, 31920707; Mata-Greenwood, 2017, 28620317}. As you will notice in our published reports, endothelial cells have different morphology and form very different capillary-like tubes (Figure 2G) that only stain with lectin endothelial stains (similar to those observed in adult arteries). We have added new panels in Figure 2, showing the morphology of endothelial cells (Figure 2E) and FVSCs (Figure 2F). Unlike endothelial cells, the cells from fetal carotids form a tube-like structure that stains for endothelial stain as well as myosin stain. Similarly, when fibroblasts were plated on the collagen gel – they did not form any tube-like structure and failed to stain by lectin or myosin. Thus, we label these cells as vascular stem cells. Moreover, this is the first report to show that a different population of cells exists in fetal arteries which are sensitive to B-Raf inhibition. Also, the research is in agreement with the published findings that B-Raf knockout mice die in-utero because of vascular defects. We hope that following the publication of this research, investigators will further characterize these cells. This is an ongoing project in the lab, and we will continue to characterize them. Furthermore, we request that the reviewers see this report as an initiation of research regarding fetal vascular stem cells and not a final verdict on these cells. We hope the reviewers will let our report be published because it is important for the investigators in the scientific community to be aware of differences in cell populations in fetal versus adult vessels and sensitivity to B-Raf.

Reviewer 2 Report

This article investigates vascular stem cells from the ovine metal carotid artery. These are investigated with respect to  signalling and survival. B-raf is unregulated and via STAT3 confers survival signals by increasing Bcl2.

Whereas the part describing B-raf's importance for Bcl2 and survival in the cell populations is OK, the characterisation of the "vascular stem cells" is tenuous. The structures that are indicated as vascular sprouts are not well defined. In Fig 1, do they have a lumen? In fig 2, I see no clear distinct delineation between the red (muscle) and green (endothelial cells) staining. In Fig 3, most staining overlap. VEGF is not an endothelial marker and shouldn't nanog stain nuclei?

In summary, this manuscript requires additional data on the identification of endothelial cells to make a convincing story.

Author Response

The importance of this paper is to demonstrate a different population of cells in fetus vs. adult arteries. Also, comparative analysis shows that fetal cells have the ability to form de-novo tube-like structures that stain for both endothelial and muscle markers. We have included an un-overlapped figure of both stainings (Figures 2A and 2B). Also, we have corrected the typo VEGF as VEGF-R, which was used as an endothelial marker. Additionally, we have previously published on endothelial cells {Goyal, 2019, 31920707;Mata-Greenwood, 2017, 28620317} and as shown in these publications, these cells have a different morphology as compared to fetal vascular stem cells. As mentioned in these publications, endothelial cells have a cobblestone appearance, and their tube formation in 3-D culture is significantly different than observed by the fetal VSCs obtained in these studies. The endothelial cells form tube-like structures as we see in cells from adult arteries. We have included a new figure (Figure 2E, F, &G) showing endothelial morphology and their behavior on 3D culture, which is very different from the phenotype of these cells. Also, endothelial cells are not sensitive to B-Raf inhibition. We have included new data and revised the manuscript.

Round 2

Reviewer 1 Report

This reviewer would like to thank the authors for the responses and this reviewer is also in agreement with the authors with the notion that a population of stem cells is present in fetal but not adult vasculature and that this finding is indeed of great interest for the scientific community. However, without some basic initial characterization and purification this reviewer believes that one should be very careful with any assumptions made henceforth. From the initial explants what percentage of cells are deemed vascular stem cells? Do the authors believe that these stem cells are present in high enough population in the initial explant to overgrow any other cell type when in culture? Have the authors considered that embryonic cells in general may respond differently to B-Raf inhibition when compared to adult, matured cells? Again, this reviewer does not refute the existence of a different population in fetal versus adult tissue and the importance of its characterization to the scientific community, nonetheless this reviewer still believes that proper initial sorting is essential for the manuscript.

Reviewer 2 Report

no comment